# Can Parenting Practices Moderate the Relationship between Reward Sensitivity and Adolescents’ Consumption of Snacks and Sugar-Sweetened Beverages?

**DOI:** 10.3390/nu12010178

**Published:** 2020-01-08

**Authors:** Wendy Van Lippevelde, Leentje Vervoort, Jolien Vangeel, Lien Goossens

**Affiliations:** 1Department of Marketing, Innovation and Organisation, Ghent University, 9000 Ghent, Belgium; 2Department of Nutrition and Public Health, University of Agder, 4604 Kristiansand, Norway; 3Department of Developmental, Personality and Social Psychology, Ghent University, 9000 Ghent, Belgium; Leentje.Vervoort@ugent.be (L.V.); Lien.Goossens@ugent.be (L.G.); 4Department of Business studies and Business Administration, Karel De Grote University College, 2000 Antwerp, Belgium; Jolien.Vangeel@kdg.be

**Keywords:** adolescent, reward sensitivity, parents, nutrition, environment, snacks, sugar-sweetened beverages

## Abstract

Background: Reward sensitivity has been associated with adolescents’ intake of unhealthy snacks and sugar-sweetened beverages. However, so far, there are no studies published describing the impact of parenting practices on this relationship. The present study will, therefore, investigate whether food parenting practices can moderate the association between reward sensitivity and diet intakes. Method: A cross-sectional research study was conducted among 14- to 16-year old Flemish adolescents (*n* = 867, age 14.7 ± 0.8 y, 48.1% boys) and a subset of their parents (*n* = 131), collecting data on daily intakes, reward sensitivity, and food parenting practices. Linear regression was used to assess the moderation effect of parenting practices (both adolescent- and parent-reported) on the relationship between reward sensitivity, and diet using SPSS 25.0. Results: In the main analysis (adolescent-reported), no significant moderation effects were found for parenting practices on the relationship between reward sensitivity and diet. However, the sensitivity analysis (parent-reported) showed a moderation effect for health-reducing parenting practices on the association between reward sensitivity and unhealthy snack intake (β = 0.297, 95% CI = 0.062, 0.531, *p* = 0.01). Conclusion: Given the difference in the effect of parenting practices between the adolescent- and parent-reported data, our inconclusive findings warrant more research in larger adolescent-parent dyad samples.

## 1. Introduction

Adolescents’ eating pattern is characterized by a high intake of energy-dense snacks and sugar-sweetened beverages (SSBs) [1,2]. Research demonstrates that over-consumption of these unhealthy snacks and SSBs is predictive of obesity and other health problems [3,4,5]. In the current obesogenic environment, there is an abundance of palatable foods and sugar-sweetened drinks [6,7]. Therefore, it is important to increase our knowledge on which factors explain adolescents’ increased intake of unhealthy snacks and SSBs.

At the individual level, some adolescents seem to be more vulnerable to increase their food intake when they are faced with highly palatable (or reinforcing) foods like energy-dense snacks or SSBs [8]. More specifically, recent research seems to indicate that high levels of the temperamental trait of reward sensitivity may increase youngsters’ vulnerability to over-consumption of palatable foods and SSBs [8]. Reward sensitivity can be defined as the proneness to detect signals of reward in the environment and to experience positive effects in rewarding situations [9,10]. More specifically, it can be seen as the reflection of interindividual differences in the sensitivity of the behavioral activation system (BAS-system). An individual with a strongly reactive approach system (i.e., a sensitive behavioral activation system), will be highly sensitive to reward or to cues that signal reward [11]. Previous studies related higher levels of reward sensitivity to an increased experience of food cravings, more emotional eating, and an increased risk of being overweight in children and adults [9,12,13]. However, the study by De Cock and colleagues [8] was the first to demonstrate that adolescents’ self-reported reward sensitivity was positively associated with daily unhealthy snack intake and the intake of SSBs (especially in girls). 

Importantly, adolescents’ food choices may not only be explained by individual characteristics but also by environmental factors [14]. Decades of research have shown that the home environment plays an important role in shaping children’s eating habits [15,16,17,18]. In general, parenting practices refer to the behaviors or actions (intentional or unintentional) performed by parents for child-rearing purposes that influence their child’s attitudes, behaviors, or beliefs [15]. Specific practices that are used within the context of feeding are called food parenting practices. Such practices may be of more or less controlling nature. Controlling practices are regarded as attempts by parents to regulate or influence the child’s eating behavior by pressuring the child to eat certain foods or to empty his plate and by restricting the intake of energy-dense food [16]. Previous research shows that the use of controlling food parenting practices such as pressure to eat and restriction are related to more negative eating and weight-related outcomes because they disrupt children’s innate self-regulation mechanisms [17]. Therefore, these parenting practices can be considered health-reducing. For example, Birch and colleagues [17] found that maternal restriction of the child’s food intake at five years predicted more eating in the absence of hunger when the child was between seven and nine years old, especially in overweight children. Less controlling practices like monitoring, encouragement, as well as modeling of healthy eating behavior and making healthy food available at home are considered more adaptive and health-promoting since they appear to be associated with more positive eating- and weight-related outcomes. For example, parental encouragement and modeling of healthy eating have been associated with healthy eating in children [15,16,18].

Past research often focuses on how food parenting practices influence dietary intake in toddlers and elementary school-aged children, and to date, far less studies have been carried out in samples of adolescents. Although adolescence is a developmental period that is characterized by an increased strive for autonomy, at the same time, adolescent boys and girls are still dependent on their parents, and the quality of the parent-child relationship remains important for determining food intake [19]. In adolescent samples, some European studies examined the association between more general parenting styles, such as authoritative, authoritarian, indulgent and neglectful parenting, and consumption behavior [20,21]. Regarding the role of more specific food parenting practices, Loth and colleagues [22] found in their population-based study in the US that the use of controlling food-related parenting practices such as pressure to eat and the restriction is related to adolescents’ weight status. In a recent study, it was also demonstrated that more use of controlling parenting practices, where rules and limits are put forward to avoid junk food and SSBs, seems associated with more consumption of junk foods and SSBs in the US adolescents [23]. As it is the case in younger children, some studies in adolescent samples also concluded that having family meals on a regular basis, and modeling and encouragement of healthful eating by parents are more health-promoting as they are associated with less fast food consumption and more healthy eating in their adolescents [24,25]. However, more research is needed to unravel whether and how food parenting practices are implicated in determining adolescents’ eating behavior in this challenging developmental period. 

In recent developmental models such as the goodness of fit model [26], it is generally assumed that the interaction between a children’s and adolescents’ temperament and its environment is one of the core mechanisms leading to (mal)adaptive behavior. When the environment is able to recognize and accommodate a child’s and adolescent’s temperament, development is more likely to be prosperous than when there is no compatibility between a child’s and adolescent’s temperament and the environment in which it grows up. In line with this view, unhealthy eating and drinking behavior (such as increased intake of energy-dense snacks and SSBs) in adolescents may be seen as a result of the interaction between adolescents’ level of reward sensitivity and their parents’ use of certain food parenting practices. Therefore, in the present study, we aim to investigate whether food parenting practices moderate the negative impact of reward sensitivity on the intake of snacks and SSBs in adolescents. More specifically, we hypothesize that health-promoting (i.e., availability of healthy products) parenting practices attenuate this relationship, whereas health-reducing (i.e., coercive control) parenting practices strengthen this negative relationship.

## 2. Materials and Methods

### 2.1. Study Procedure

The present study was a secondary data analysis on data from the REWARD project (http://www.rewardstudy.be). The overall aim of this project was to provide evidence for a new public health framework to improve the eating patterns of children and adolescents by focusing on individual differences in reward sensitivity. Data were collected between September and December 2013 in a representative sample of adolescents recruited in the third and fourth grades of 20 secondary schools in Flanders (Belgium). Additionally, parents of the participating adolescents were also invited to fill out the parent questionnaire (one parent per adolescent). Parents’ passive consent and adolescents’ active consent were obtained prior to participation of the adolescents, and parents’ active consent was given for their own participation. Of the 1210 selected adolescents, 6% were absent or not allowed to participate, and 3% returned a questionnaire of unsatisfactory quality (defined as more than 33% of the questions not completed or straight-lining responses) for further use. More detailed information on the study procedure and used materials can be found elsewhere [8,27]. The study protocol was approved by the Ethics Committee of the Ghent University Hospital and performed in accordance with the ethical standards laid down in Belgian national laws and the Declaration of Helsinki.

### 2.2. Measures

The adolescent questionnaire assessed socio-demographics, reward sensitivity, perceived parenting practices, and snack and sugar-sweetened beverage intake. The parent questionnaire also assessed socio-demographics and applied parenting practices.

### 2.3. Adolescents’ Intake of Sugar-Sweetened Beverages (SSB) and (Un)Healthy Snacks

Snack food and sugar-sweetened beverage (SSB) consumption was assessed using a Food Frequency Questionnaire (FFQ) for adolescents [27], based on the validated diet quality index for children [28]. The six categories used were: 0 = never or seldom; 1 = 1–3 days/month; 2 = 1 days/week; 3 = 2–4 days/week; 4 = 5–6 days/week; 5 = every day. These categories were recoded into the frequency of consumption per week: 0 = never or seldom; 0.5 = 1–3 days/month; 1 = 1 days/week; 3 = 2–4 days/week; 5.5 = 5–6 days/week; 7 = every day [28]. The FFQ probes for the consumption of snacks and beverages within a reference period of one month. The 28 snack items were: chocolate and pralines, candy bars, candy, dry cookies, other cookies such as chocolate cookies, breakfast rolls, pastries, breakfast cereals, unsweetened yogurt, sweetened yogurt, pudding, mousses, ice-cream, popsicles, dried fruit, fruit, raw vegetables, nuts and seeds, sandwiches with sweet or savory spread, cheese or meat cubes, crisps and similar products, other savory snacks such as breadsticks, sausage/cheese rolls, and pizza, other fried snacks such as spring rolls and cheese croquettes, French fries, kebab, hamburgers, and pasta cups. Included sugar-sweetened beverage items were energy drinks, sports drinks, soft drinks. Snacks were classified as either healthy or unhealthy using the UK Ofcom nutrient profiling model [29]. Following this scoring system, the FFQ snack items chocolate and pralines, candy bars, candy, dry cookies, other cookies such as chocolate cookies, breakfast rolls, pastries, breakfast cereals, pudding, mousses, ice-cream, popsicles, cheese or meat cubes, crisps and similar products, other savory snacks such as bread sticks, sausage/cheese rolls and pizza, other fried snacks such as spring rolls and cheese croquettes, French fries, hamburgers were considered to be unhealthy (20) and the other FFQ snack items healthy (8). Depending on the item, 4–6 portion-size categories (in g or mL) were provided together with a list of common standard measures as examples of the quantity of consumption for the most frequently consumed food items by adolescents within that category, as reported in the European HELENA study [8]. The average (in g or mL) of the portion ranges and the lowest/highest category ±0.67 was used to calculate the quantity of the consumption. For instance, for candy the following portion sizes were given 9 g or less (i.e., 6 g), 10–34 g (i.e., 22 g), 35–59 g (i.e., 47 g), 60–84 g (i.e., 72 g), 85–109 g (i.e., 97 g), and 110 g or more (i.e., 183.3 g), together with the following examples of portions—one small bag of M&M’s = 45 g and one wine gum = 4 g. The daily intake of each FFQ item was obtained by multiplying the frequency of consumption per week with the quantity of consumption (g or mL) divided by 7. The overall daily intakes per item were then summed to obtain the daily intakes of unhealthy snacks (g), healthy snacks (g), and SSBs (mL). Zero imputation (i.e., assumption of no consumption) was used for food items that were left blank. We had complete cases for 87% of the snack items, 8.2% had a missing for one item, and less than five percent had more than one missing for the snack items. For the SSBs, less than two percent had one or more missing items for the SSB items.

### 2.4. Adolescents’ Reward Sensitivity

Reward Sensitivity (RS) was indexed by the BAS-scale of a Dutch age-downward version one of Carver and White’s BIS/BAS scale [30]. Thirteen items are scored on a four-point scale (1 = not at all true, 2 = somewhat not true, 3 = somewhat true, 4 = all true), and summed to obtain Total RS scores. Psychometrics of this Dutch version of the BIS/BAS-scales are evaluated in Dutch and Flemish children and adolescents aged 8–18 years [30,31], evidencing that the total scale is a valid measurement of RS in children and adolescents with higher scores indicating more RS (range 13–52). The internal consistency of the total scale in the present sample was very good (Cronbach’s α = 0.82).

### 2.5. Health-Promoting and Health-Reducing Food-Parenting Practices Perceived by Adolescents and Parents

The food parenting practices were assessed separately in both adolescents (adolescent-reported) and their parents (parent-reported) by 21 items (using five-point Likert scales) from the adolescent and parent version of the Child Feeding Questionnaire [32] and the Comprehensive Feeding Practices Questionnaire [33]. Both questionnaires have been tested for their validity and reliability. Measured constructs were modeling (three items), monitoring (seven items), encouragement (five items), availability (two items), restriction (one item), and pressure to eat (three items). The items of the first four constructs were combined—based on both theoretical and empirical grounds—into the overall construct “health-promoting parenting practices”. The items of the latter two constructs were combined—based on theory and empirical grounds—into the overall construct “health-reducing parenting practices”.

### 2.6. Potential Confounders

The following a-priori defined covariables were considered as potential confounders due to a known association with diet [34]). Adolescents’ age, gender, family situation (traditional = the two parents living together versus non-traditional), education type (general versus vocational; technical versus vocational), and parents’ age, gender, and socio-economic status (based on education of mother and father), and meal patterns (daily breakfast, lunch, and dinner versus not daily breakfast, lunch, and dinner) were assessed via self-reported questionnaires.

### 2.7. Statistical Analyses

#### 2.7.1. Main Analysis

Descriptive statistics were produced (see Table 1). Analyses were conducted on complete cases, and therefore only included participants who had valid measures for reward sensitivity, covariables, parenting practices, and diet outcomes. To investigate the moderation effects of both health-promoting and health-reducing parental practices on the association between reward sensitivity and SSB and (un)healthy snack intake in adolescents, a set of hierarchical linear regression models were conducted using SPSS version 25.0. First, an unadjusted model without the interaction term BAS total*parenting practices (model 1); second, an unadjusted model with the interaction term BAS total*parenting practices (model 2); third, a model adjusting for adolescents’ age (continuous), gender (dichotomous), family situation (dichotomous), and education type (two dummies were created) (model 3); and finally a model that included the covariables adjusted for in model 3 plus additional adjustment for breakfast, lunch, and dinner habits (model 4).

#### 2.7.2. Sensitivity Analysis

Sensitivity analyses were performed to evaluate the robustness of the findings (see Appendix A, Table A1, Table A2, Table A3, Table A4, Table A5 and Table A6) by rerunning the analyses first including all cases, regardless of missing values; second with ≥3 SD outlier exclusion for the dietary intake variables, and third using the (health-promoting and health-reducing) parenting practices reported by the parents. Generally, findings were similar under these conditions and therefore will not be commented on further, apart from the significant moderation effect in the parent sample.

## 3. Results

### 3.1. Description of Sample

One-thousand one-hundred and four adolescents (and 158 parents) completed the questionnaires. The complete-cases analysis sample was comprised of 867 adolescents (and 131 parents). Table 1 presents characteristics of the adolescent sample. The mean age of the participants was 14.7; there were slightly fewer boys (48.1%). Most adolescents came from traditional families, and most participating adolescents were in general education. The participating parents had a mean age of 44.7 ± 3.7, were mostly mothers (81.7%), and from higher SES families (76.3%). Mean health-promoting and health-reducing parenting practices were respectively 3.6 ± 0.4 and 2.7 ± 0.7.

### 3.2. Moderation Effect of Health-Promoting and Health-Reducing Parenting Practices

Table 2 and Table 3 show the associations between reward sensitivity, the (adolescent-reported) parenting practices, and covariables, and the included dietary measures (i.e., daily intake of SSB, healthy and unhealthy snacks). Despite several significant main associations between reward sensitivity and parenting practices and SSB, healthy and unhealthy snack consumption (also reported in [27]), no significant moderation effects were found for health-promoting or health-reducing parenting practices on the relationship between reward sensitivity and the diet measures.

In the sensitivity analyses, we did find a moderation effect for parent-reported health-reducing parenting practices on the relationship between reward sensitivity and unhealthy snack intake. Figure 1 presents this interaction effect and shows a more unhealthy snack intake when using more health-reducing parenting practices compared to a lower intake of unhealthy snacks when using less health-reducing parenting practices among highly reward sensitive adolescents. Figure 1 shows no difference between the use of health-reducing parenting practices among adolescents with lower reward sensitivity.

## 4. Discussion

The present study investigated whether the influence of reward sensitivity, a known determinant of snack and SSB intake in adolescents [8], might be affected by either health-promoting or health-reducing food parenting practices. Following the goodness-of-fit [26] model, suggesting that the interaction between an adolescent’s temperament and its environment is one of the core mechanisms leading to a (mal)adaptive behavior, we assumed that the association between reward sensitivity and unhealthy eating and drinking behavior might be attenuated by health-promoting parenting practices while being enhanced by health-reducing parenting practices. Although the data reported by the adolescents partly supported the role of both reward sensitivity and parenting practices separately, they did not provide evidence for their interactive role in unhealthy and healthy snack intake and SSB consumption in adolescents. Based on the analysis of the complete case, data suggested that higher reward sensitivity was associated with a higher intake of healthy and unhealthy snacks and higher consumption of SSB. This finding is in line with previous research where it was found that higher levels of reward sensitivity were associated with eating- and weight-related outcomes such as the increased experience of food cravings, more emotional eating, and an increased risk for being overweight in children and adults [9,12,13].

However, when diet outliers were removed, these effects were no longer significant. In contrast, the effects of parenting practices were more robust and survived all sensitivity analyses. Increased use of health-promoting parenting practices was associated with reduced SSB consumption and increased healthy snack intake, while increased use of health-reducing parenting practices was associated with increased unhealthy snack intake. These results are in line with previous studies [22,23] and seem to demonstrate that according to the adolescents themselves, parental variables (in this case, parenting practices) still play an important role in explaining adolescents’ eating and drinking behavior. Although these results are only cross-sectional, they do seem to indicate that during the developmental stage of adolescence, where increased autonomy is required, working with parents to address their adolescents’ eating behavior may still be an effective intervention strategy.

Parent-reported data on parenting practices; however, did provide evidence for their interactive role with reward sensitivity in explaining adolescents’ unhealthy snack intake. More specifically, results demonstrate that health-reducing parenting practices strengthen the association between adolescents’ reward sensitivity and unhealthy snack intake. In other words, in line with the assumptions of the goodness of fit model [26], unhealthy eating behavior may be explained by a ‘poorness of fit’ between adolescent’s vulnerable temperament and their parents’ food-related parenting practices. The other two outcomes, SSB consumption, and healthy snack intake, were not determined by parenting practices (either alone or in interaction with reward sensitivity).

The present study has several strengths. First of all, the divergence of the results obtained through adolescent and parent report point to the relevance of multi-informant assessment. Although considered best practice in developmental psychopathology research [35], this approach is less often used in nutrition research. However, both parents and adolescents might experience, and thus, report on a certain concept (like parenting) differently, which might at the one sight create measurement error but, on the other side, open up a broader view on its role in determining behavior. Previous studies, including both child- and parent-report in investigating parental influence, showed stronger associations with child eating behavior for the child perception than parent perception [36]. In the present study, we were able to collect both parents- and adolescent-reported data on food parenting practices in a small subsample of our participants, giving a first impression on its role in snack and SSB intake in adolescents. The results based on parent-report; however, need replication. Future research might aim to collect data in a large and representative sample of adolescent-parent dyads. In order to get a more comprehensive view of how parents influence their adolescents’ eating behavior, other methodologies are required. For example, observational methods can provide information about the verbal and non-verbal context of behavioral dynamics during mealtimes, e.g., [37].

A large sample is necessary to have enough power to detect small effects. We assumed that the effect sizes of the determinants under study would be small because eating behavior has multiple determinants that act together on different levels [1,14]. Therefore, we collected data in a large sample. In order to be able to generalize our results to the Flemish population of adolescents between 14 and 16 years, we ensured that our sample was representative. In addition to its multi-informant assessment of parenting practices and its large and representative sample, another strength of the present study is its use of validated instruments to assess reward sensitivity [30] and snack and SSB intake [27]. 

Some limitations have to be noted, as well. First, parenting practices were assessed through a purpose-built questionnaire, using a selection of relevant items borrowed from several instruments on food parenting practices, being the Child Feeding Questionnaire [32], and the Comprehensive Feeding Practices Questionnaire [33]. Future research might benefit from using the validated instruments themselves. Moreover, as argued by other researchers, different concepts are often used to study food parenting practices, and future research should strive for more consensus on how to operationalize and measure different practices [38]. Next, only a limited subsample of parents reported on their parenting practices, which call into question the representativeness of this subsample in comparison to the entire sample and the population. Furthermore, because of the cross-sectional nature of the present study, no conclusions can be drawn regarding the direction of the associations and causality. In order to examine whether reward sensitivity, parenting practices, and their interaction predict unhealthy snacking and the intake of SSB, longitudinal and experimental designs are needed. Moreover, future research may investigate whether other environmental factors interact with reward sensitivity to explain adolescent’s intake of snacks and SSB [11]. For example, previous research found evidence for the role of peer consumption and the school environment in explaining adolescents’ snack and soft drink consumption [39]. Studying how individual (intrapersonal) characteristics interact with different environmental factors may result in a more comprehensive explanation of adolescents’ eating/snacking behavior and may lead to more tailored intervention efforts that target factors at multiple levels of influence.

To conclude, the present study investigated whether health-promoting or health-reducing food parenting practices could impact the relationship between reward sensitivity and dietary intake in adolescents. The main analysis on the adolescent-reported data did not show any moderating effect of food parenting practices. This finding indicates that, although reward sensitivity and parenting practices were uniquely associated with dietary intake, they do not seem to interact based on adolescent-reported data. Other—for example, observational—measures and multiple informants are needed to investigate further whether and how parenting practices impact the relation between temperamental traits and adolescents’ dietary intake. Especially since the sensitivity analysis using the small sample of parent-reported data showed an interactive role with reward sensitivity in explaining adolescents’ unhealthy snack intake. This finding indicates that health-reducing parenting practices seemed to strengthen the association between adolescents’ reward sensitivity and unhealthy snack intake. Nevertheless, more longitudinal and intervention research in larger samples of adolescent-parent dyads is necessary to further investigate the impact of both health-promoting and health-reducing parenting practices on the relationship between reward sensitivity and diet intakes of adolescents.

## Figures and Tables

**Figure 1 nutrients-12-00178-f001:**
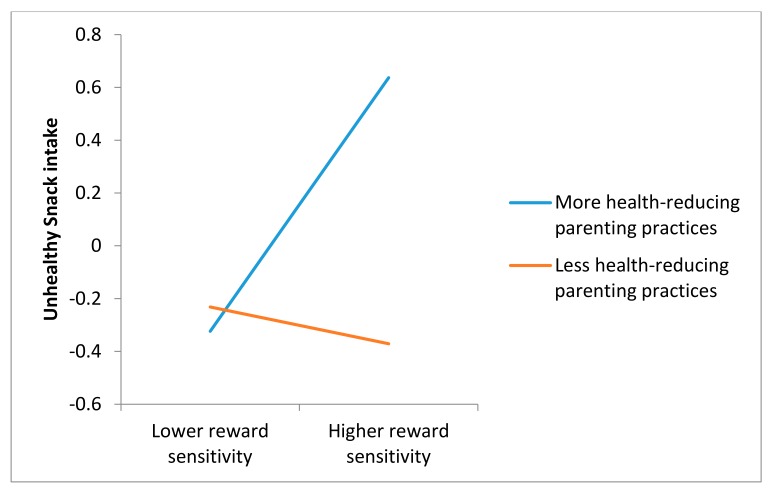
The moderation effect of Reward Sensitivity and Health-reducing parenting practices on the intake of unhealthy snacks.

**Table 1 nutrients-12-00178-t001:** Participant characteristics* in those eligible and included and those eligible but excluded due to missing data on at least one independent or co-variable.

	Eligible & Included *n* = 867 Mean ± SD or %	Eligible but Missing data on Independent or Co-Variables *n* = 237 Mean ± SD or %	*p*-Value
Baseline Characteristics			
Age	14.7 ± 0.8	14.8 ± 0.9	0.2
Boys	48.1	58.2	<0.001
Family type			0.9
Traditional	70.9	70.5	
Other	29.1	29.5	
School type			<0.001
General	48.9	35.9	
Technical	34.4	32.5	
Vocational	16.7	31.6	
Independent Variables			
BAS total (1–52)	31.9 ± 6.4	30.6 ± 6.8	0.02
Health-promoting parenting practices (1–5)	3.11 ± 0.65	3.06 ± 0.65	0.3
Health-reducing parenting practices (1–5)	2.91 ± 0.81	2.89 ± 0.73	0.7
Dietary Outcomes			
Daily intake of sugar–sweetened beverages [in mL]	361.8 ± 398	427.4 ± 512.5	0.07
Daily intake of healthy snacks [in g]	182.7 ± 216.2	168.8 ± 277.2	0.4
Daily intake of unhealthy snacks [in g]	256.4 ± 287.3	280.6 ± 361.8	0.4

BAS total (Behavioral Activation System) is an indicator of Reward sensitivity, SD = standard deviation, g = grams, mL = milliliter.

**Table 2 nutrients-12-00178-t002:** Associations between reward sensitivity and adolescents’ intake of sugar-sweetened beverages, healthy and unhealthy snacks, and the moderating role of (adolescent-reported) health-promoting parenting practices (*n* = 867).

	Model 1		Model 2		Model 3		Model 4	
	B (95% CI)	*p*-Value	B (95% CI)	*p*-Value	B (95% CI)	*p*-Value	B (95% CI)	*p*-Value
Outcome: Sugar-sweetened beverages	Adjusted R^2^ = 0.046		Adjusted R^2^ = 0.046		Adjusted R^2^ = 0.122		Adjusted R^2^ = 0.125	
BAS total	0.103 (0.038, 0.168)	0.002	0.109 (0.043, 0.175)	0.001	0.97 (0.033, 0.162)	0.003	0.096 (0.031, 0.160)	0.004
Health-promoting parenting practices	−0.200 (−0.266, −0.135)	<0.001	−0.204 (−0.270, −0.139)	<0.001	−0.152 (−0.217, −0.087)	<0.001	−0.144 (−0.210, −0.079)	<0.001
BAS total * Health-promoting parenting practices			0.032 (−0.029, 0.094)	0.3	0.021 (−0.038, 0.080)	0.5	0.023 (−0.037, 0.082)	0.5
Age					0.006 (−0.073, 0.086)	0.9	−0.003 (−0.083, 0.077)	0.9
Gender [Ref: boy]					−0.432 (−0.558, −0.305)	<0.001	−0.452 (−0.580, −0.323)	<0.001
Family type [Ref: not traditional]					−0.067 (−0.207, 0.073)	0.3	−0.050 (−0.190, 0.091)	0.5
School type General [Ref: Vocational]					−0.427 (−0.617, −0.238)	<0.001	−0.409 (−0.600, −0.218)	<0.001
School type Technical [Ref: Vocational]					−0.171 (−0.362, 0.021)	0.08	−0.167 (−0.359, 0.024)	0.09
Breakfast [Ref: Not daily]							−0.158 (−0.294, −0.023)	0.2
Lunch [Ref: Not daily]							−0.022 (−0.179, 0.136)	0.8
Dinner [Ref: Not daily]							0.087 (−0.127, 0.301)	0.4
Outcome: Healthy Snack intake	Adjusted R^2^ = 0.018		Adjusted R^2^ = 0.017		Adjusted R^2^ = 0.021		Adjusted R^2^ = 0.020	
BAS total ^#^	0.070 (0.004, 0.136)	0.04	0.069 (0.002, 0.136)	0.04	0.067 (0.000, 0.135)	0.05	0.067 (−0.001, 0.136)	0.05
health-promoting parenting practices	0.120 (0.054, 0.186)	<0.001	0.121 (0.054, 0.187)	<0.001	0.111 (0.043, 0.180)	0.001	0.107 (0.038, 0.176)	0.003
BAS total * health-promoting parenting practices			−0.005 (−0.067, 0.058)	0.9	−0.002 (−0.064, 0.061)	1.0	−0.001 (−0.064, 0.062)	1.0
Age					−0.001 (−0.085, 0.082)	1.0	0.002 (−0.082, 0.087)	1.0
Gender [Ref: boy]					0.118 (−0.015, 0.252)	0.08	0.126 (−0.010, 0.262)	0.07
Family type [Ref: not traditional]					−0.029 (−0.177, 0.119)	0.7	−0.034 (−0.183, 0.114)	0.6
School type General [Ref: Vocational]					0.080 (−0.120, 0.280)	0.4	0.071 (−0.131, 0.273)	0.5
School type Technical [Ref: Vocational]					−0.083 (−0.285, 0.119)	0.4	−0.084 (−0.287, 0.119)	0.4
Breakfast [Ref: Not daily]							0.082 (−0.061, 0.226)	0.3
Lunch [Ref: Not daily]							−0.043 (−0.209, 0.124)	0.6
Dinner [Ref: Not daily]							−0.002 (−0.229, 0.225)	1.0
Outcome: Unhealthy Snack intake	Adjusted R^2^ = 0.017		Adjusted R^2^ = 0.017		Adjusted R^2^ = 0.021		Adjusted R^2^ = 0.020	
BAS total	0.147 (0.081, 0.213)	<0.001	0.069 (0.085,0.220)	<0.001	0.146 (0.079, 0.213)	<0.001	0.143 (0.076, 0.210)	<0.001
Health-promoting parenting practices	−0.056 (−0.122, 0.010)	0.1	−0.060 (−0.126, 0.007)	0.08	−0.030 (0.097, 0.037)	0.4	−0.029 (−0.097, 0.038)	0.4
BAS total * Health-promoting parenting practices			0.030 (−0.032, 0.092)	0.3	0.025 (−0.036, 0.086)	0.4	0.024 (−0.037, 0.086)	0.4
Age					−0.002 (−0.084, 0.080)	1.0	−0.005 (−0.088, 0.078)	0.9
Gender [Ref: boy]					−0.328 (−0.459, −0.196)	<0.001	−0.335 (−0.468, −0.201)	<0.001
Family type [Refe: not traditional]					0.097 (−0.048, 0.242)	0.2	0.098 (−0.048, 0.244)	0.2
School type General [Ref: Vocational]					−0.292 (−0.488, −0.095)	0.004	−0.280 (−0.478, −0.081)	0.006
School type Technical [Ref: Vocational]					−0.167 (−0.365, 0.031)	0.1	−0.160 (−0.359, 0.039)	0.1
Breakfast [Ref: Not daily]							0.011 (−0.130, 0.152)	0.8
Lunch [Ref: Not daily]							−0.041 (−0.205, 0.123)	0.6
Dinner [Ref: Not daily]							−0.097 (−0.320, 0.125)	0.4

* Results in the table are for complete cases (*n* = 867) with unstandardized B values; Model 1 includes the main independent variables (BAS total, Health-promoting parenting practices); Model 2 includes the main independent variables (BAS total, Health-promoting parenting practices) and the interaction; Model 3 includes adjustments for age, gender, family type (traditional versus other), and school type (general, technical, vocational); Model 4 includes model 3 adjustments plus additional adjustments for breakfast, lunch, and dinner patterns. ^#^ BAS total (Behavioral Activation System) is an indicator of Reward sensitivity.

**Table 3 nutrients-12-00178-t003:** Associations between reward sensitivity and adolescents’ intake of sugar-sweetened beverages, healthy and unhealthy snacks, and the moderating role of (adolescent-reported) health-reducing parenting practices (*n* = 867).

	Model 1		Model 2		Model 3		Model 4	
	B (95% CI)	*p*-Value	B (95% CI)	*p*-Value	B (95% CI)	*p*-Value	B (95% CI)	*p*-Value
Outcome: Sugar-sweetened beverages	Adjusted R^2^ = 0.006		Adjusted R^2^ = 0.005		Adjusted R^2^ = 0.100		Adjusted R^2^ = 0.106	
BAS total ^#^	0.092 (0.025, 0.160)	0.007	0.092 (0.025, 0.160)	0.007	0.080 (0.015, 0.145)	0.02	0.079 (0.014, 0.144)	0.02
Health-reducing parenting practices	−0.011 (−0.078, 0.056)	0.8	−0.011 (−0.079, 0.056)	0.7	0.001 (−0.064, 0.066)	1.0	−0.005 (−0.070, 0.060)	0.9
BAS total * Health-reducing parenting practices			0.006 (−0.056, 0.067)	0.9	−0.001 (−0.060, 0.058)	1.0	0.001 (−0.058, 0.059)	1.0
Age					0.016 (−0.065, 0.096)	0.7	0.003 (−0.078, 0.084)	0.9
Gender [Ref: boy]					−0.453 (−0.581, −0.325)	<0.001	−0.476 (−0.605, −0.346)	<0.001
Family type [Ref: not traditional]					−0.093 (−0.234, 0.049)	0.2	−0.072 (−0.214, 0.070)	0.3
School type General [Ref: Vocational]					−0.509 (−0.697, −0.320)	<0.001	−0.480 (−0.670, −0.289)	<0.001
School type Technical [Ref: Vocational]					−0.238 (−0.429, −0.046)	0.02	−0.227 (−0.419, −0.036)	0.02
Breakfast [Ref: Not daily]							−0.191 (−0.328, −0.055)	0.006
Lunch [Ref: Not daily]							−0.007 (−0.166, 0.153)	0.9
Dinner [Ref: Not daily]							0.068 (−0.149, 0.285)	0.5
Outcome: Healthy Snack intake	Adjusted R^2^ = 0.004		Adjusted R^2^ = 0.005		Adjusted R^2^ = 0.012		Adjusted R^2^ = 0.011	
BAS total	0.079 (0.011, 0.146)	0.02	0.078 (0.010, 0.145)	0.02	0.078 (0.010, 0.146)	0.03	0.077 (0.009, 0.146)	0.03
health-reducing parenting practices	−0.007 (−0.74, 0.061)	0.8	−0.003 (−0.071, 0.064)	0.9	−0.005 (−0.073, 0.063)	0.9	−0.002 (−0.070, 0.067)	1.0
BAS total * health-reducing parenting practices			−0.043 (−0.105, 0.018)	0.2	−0.044 (−0.105, 0.018)	0.2	−0.044 (−0.106, 0.017)	0.2
Age					−0.012 (−0.096, 0.073)	0.8	−0.006 (−0.091, 0.080)	0.9
Gender [Ref: boy]					0.134 (0.000, 0.269)	0.05	0.144 (0.008, 0.280)	0.04
Family type [Refe: not traditional]					−0.018 (−0.166, 0.131)	0.8	−0.026 (−0.175, 0.123)	0.7
School type General [Ref: Vocational]					0.134 (−0.063, 0.332)	0.2	0.118 (−0.082, 0.318)	0.2
School type Technical [Ref: Vocational]					−0.038 (−0.239, 0.162)	0.7	−0.044 (−0.245, 0.158)	0.7
Breakfast [Ref: Not daily]							0.108 (−0.036, 0.251)	0.1
Lunch [Refe: Not daily]							−0.050 (−0.218, 0.117)	0.6
Dinner [Ref: Not daily]							0.009 (−0.219, 0.237)	0.9
Outcome: Unhealthy Snack intake	Adjusted R^2^ = 0.024		Adjusted R^2^ = 0.027		Adjusted R^2^ = 0.067		Adjusted R^2^ = 0.064	
BAS total	0.131 (0.064, 0.198)	<0.001	0.130 (0.063, 0.196)	<0.001	0.123 (0.056, 0.189)	<0.001	0.121 (0.055, 0.188)	<0.001
Health-reducing parenting practices	0.074 (0.007, 0.141)	0.03	0.079 (0.012, 0.146)	0.02	0.089 (0.023, 0.155)	0.008	0.086 (0.020, 0.153)	0.01
BAS total * Health-reducing parenting practices			−0.059 (−0.120, 0.002)	0.06	−0.060 (−0.120, −0.001)	0.05	−0.060 (−0.120, 0.000)	0.05
Age					0.007 (−0.075, 0.089)	0.9	0.005 (−0.078, 0.088)	0.9
Gender [Ref: boy]					−0.346 (−0.476, −0.215)	<0.001	−0.349 (−0.481, −0.216)	<0.001
Family type [Ref: not traditional]					0.080 (−0.064, 0.224)	0.3	0.079 (−0.066, 0.225)	0.3
School type General [Ref: Vocational]					−0.305 (−0.497, −0.113)	0.002	−0.298 (−0.493, −0.103)	0.003
School type Technical [Ref: Vocational]					−0.192 (−0.387, 0.003)	0.05	−0.187 (−0.383, 0.009)	0.06
Breakfast [Ref: Not daily]							0.017 (−0.123, 0.157)	0.8
Lunch [Ref: Not daily]							−0.023 (−0.186, 0.140)	0.8
Dinner [Ref: Not daily]							−0.077 (−0.299, 0.145)	0.5

* Results in the table are for complete cases (*n* = 867) with unstandardized B values; Model 1 includes the main independent variables (BAS total, health-reducing parenting practices); Model 2 includes the main independent variables (BAS total, health-promoting parenting practices) and the interaction term; Model 3 includes adjustments for age, gender, family type (traditional versus other), and school type (general, technical, vocational); Model 4 includes model 3 adjustments plus additional adjustment for breakfast, lunch, and dinner patterns. ^#^ BAS total (Behavioral Activation System) is an indicator of Reward sensitivity.

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
