# Peer review of "Can Parenting Practices Moderate the Relationship between Reward Sensitivity and Adolescents’ Consumption of Snacks and Sugar-Sweetened Beverages?"

_nutrients, 2020, doi:10.3390/nu12010178_

Round 1
Reviewer 1 Report
Can parenting practices moderate the relationship between reward sensitivity and adolescents’ consumption of snacks and sugar-sweetened beverages?
This paper is of quite broad interest in child feeding, nutrition and parenting domains of research. It draws on data from a well-known, large scale food and nutrition study that utilised validated tools for dietary assessment. This secondary analysis of data from the REWARD study remain of interest despite the data being 6 years old, given that not a lot of subsequent work in the field has been conducted, particularly in the area of reward sensitivity. I hope the following suggestions can be used by authors to strengthen the paper.
Abstract
Line 14: The following sentence needs rewording –poor grammar affects the interpretation and impact. However, no studies investigated the impact of parents on this relationship yet.
Results: I would like to have seen the effect size or analysis result reported for parents.
Conclusion: State which aspect of ‘parents’ warrants more focus, and reframe final sentence to be more instructive, to recommend direction for this future research.
Introduction
Use words instead of numeral up to ‘nine’ then numeral from ‘10’.
Lines 36 – 40. Important points but vey convoluted sentence. Please re-word for better impact.
Lines 40 – 42: This sentence/point needs a reference.
Lines 47 – 50: Very important to the reader to understand the multiple explanatory points here – could be more clearly described, perhaps as two sentences?
Lines 48 – 53: Again, important points are contained in a long convoluted sentence, and grammar could be improved.
Line 55: Decades of research ……. requires substantiation with referencing
Lines 67 – 70: Grammar could be improved for better impact and comprehension.
Lines 75 – 92: This section is very instructive and useful context for the paper. Only suggestion is to avoid mixing terminology – what does ‘youngster’ mean here – child or adolescent. Would use adolescent or ‘older children’ but suggest being consistent.
Lines 93 – 97: Is this across children and adolescents?
Lines 97 – 98: In line with this view,….. suggest rewording to show how the model has been applied to the adolescent dietary intake and food environment ie. be more specific.
Materials and Methods
As the data was collected 6+ years ago, I suggest being clear if this is secondary analysis of data collected for the REWARD study, for transparency.
Line 112: Parents’ consent or content?
Section 2.2: I would like to see the name of the tools and whether validated for those most relevant to this analysis here, even though the primary study was fully described elsewhere.
Line 117: Section 2.2: It isn’t clear whether the adolescents and parents were matched (dyads). From the numbers of participants I would assume not, but it is unclear and in sections of both the methods and the results, it would be easy to assume that results being reported are for adolescents and their parents. Please clarify this throughout.
Line 140: Referring to the line ‘The daily intake of each FFQ item was obtained by multiplying the frequency of consumption per week with the quantity of consumption (g or ml) divided by 7.’
If the FFQ is per month, how is daily intake calculated by dividing by 7? Also, were portion categories a range of grams or mls or a single figure (as per lines 126) ? More detail on these calculations would be useful to justify an extrapolations or assumptions.
Paper states on Line 144: Zero imputation (i.e., assumption of no consumption) was used for food items that were left blank. Then on Line 171: Analyses were conducted on complete cases, and therefore only included participants who had valid measures for reward sensitivity, covariables, parenting practices and diet outcomes. Therefore, if there were imputed values for diet outcomes, were they considered to be complete?
Line 154 states: The food parenting practices are assessed in both adolescents and their parents by 21 items (using five point Likert scales) from the adolescent and parent version of the Child Feeding Questionnaire [32] and the Comprehensive Feeding Practices Questionnaire [33].
In reading the results, it is very hard to tell if the Health promoting and Health reducing parenting practices are adolescent reported or parent reported, and if parent reported how they are matched to adolescent. This may have been described in another paper, but is important to include here for context. This point also relates to Tables 2 and 3.
Lines 156-157: Please be explicit as to whether these have been validated and are reliable for the adolescent population with similar demographics.
Tables: BAS and other abbreviations in Tables need to be included in footnotes or legend so Tables stand alone from text.
Discussion: Lines 224 – 245: Does this relate to adolescent reported health promoting or health reducing food parenting practices, and from Line 245 on to parent reported? Very difficult to further review this paper until this is clarified and the food parenting practice reporting is differentiated.
Line 254: I agree with this statement but in the current format, it is really difficult for the reader to differentiate results obtained through adolescent and parent report. Please clarify throughout.
Line 278: The subsample of parents needs to be stated much more explicitly throughout, especially in methods section.
It would be preferable for a conclusion to follow the strengths and limitations to round out the paper.
Reviewer 2 Report
Thank you for the opportunity to review this manuscript. Overall the analysis is well conducted, however the conclusions should be much more tentative based on the results presented.
The main finding (i.e. based on the large representative sample of children) is that there is no evidence of food parenting practices moderating the association between reward sensitivity and ssb intake, healthy snack intake, or unhealthy snack intakes. The single finding in the much smaller sample where adult variables were used that health reducing parenting practice (as assessed by the parent) has a significant moderating effect on the association between reward sensitivity and unhealthy snack intake is inadequate to conclude that 'parents can have a significant impact on the association between reward sensitivity and unhealthy dietary intakes and warrants a focus on parents in health promotion among adolescents.'
The result in the sensitivity analysis is interesting but cannot be the focus of the manuscript because of the small sample size, and the questions raised by the differences in assessment of food parenting practices between parents and their children. The difference in results is difficult to interpret.
More minor comments:
Abstract: The abstract should include quantitative results - at present, not even the sample sizes are given. I think it is incorrect to refer to an adolescent sample and an adult sample - they were both adolescent samples, but the much smaller sample included parent generated variables.
The authors have opened with a general comment about what characterises eating patterns of adolescents (Lines 34-36), but it is possible that cultural differences are relevant to the relationship between reward sensitivity and adolescents' consumption of snacks and SSBs. Therefore it is relevant to describe where studies cited are carried out (eg 3 studies cited between lines 82 - 90).
line 121: the reader is referred to 2 references for more details on the measures used in the study. Are these the correct references? They don't appear to be publications resulting from the REWARD project.
line 124: the FFQ used is referenced to results from the same project - are further details available on the method here, or on the snack food and SSB intake - this could be clarified or the citation removed.
line 126: reference needs numbering.
line 128: extra words need amending.
line 137-140: the snack items need to be identified by the exact same name as the preceding list. It would also help if the number of snack items considered unhealthy and healthy were included.
line 165 : what is considered a traditional family situation?
line 176: an explicit statement should be made of what the interaction term is.
line 180: the definition of the variable breakfast, lunch and dinner habits are not explained.
Line 190: a response rate is not given; no explanation is provided for why the sample of parents is so much smaller than that of children.
Line 220: The presentation of effect modification is very poor and difficult to make sense of. Should these be 2 straight lines, or a 2 by 2 table?
line 254: A very positive view is taken of the divergence of results according to whether adolescent or parent report is used. It should be apparent however that the models giving the divergent results are not strictly comparable - the sample is different, and different covariates are included. The possibility of measurement error might be discussed, and how food parenting practices might be objectively measured. At first glance it would appear that parents have a more health promoting view of their food parenting practices than their children.
line 282: I agree that no conclusions can be drawn about causality, but the directions of the associations are given in the analysis.
Line 301: A sample size should be given for each of the models in the sensitivity analyses - I assume that these would vary for some tables because of incomplete data.
Round 2
Reviewer 1 Report
Thank you for providing responses to reviewers that were clear and easy to see it the text and response document. The manuscript has been clarified substantially. There are several items that could be further improved or have been evidened in this version of the document. Please see below for further suggestions and comments.
Abstract
P1, L20: Suggest
……….and a subset of their parents (n=131), collecting data on……
Conclusion: Needs to be rewritten as it current restates the results. Suggest including the implications of the findings to practice, and to position and justify the need for future research.
Introduction
I believe all comments have been attended to well, and this sections reads well and provides good rationale for the study. The research gaps and aims are also now nicely defined and contextualised.
Methods
L129 – 131: I would still like to see a brief explanation of how portion size ranges/categories were selected. Were the middle portions based on standard portion or serving sizes in one country? State database/s or guideline/s used.
L131 – 135: Part of this relates to the portion ranges and part of it to the analysis. I suggest moving this to near L151.
Line 151: 8 – 18 years
Line 163: Suggest -
The food parenting practices were assessed
L182 and first line of Results: If only complete cases were used in analysis, how was a complete case defined for diet outcomes if zero imputation was used? Was there a maximum number of ‘no responses’? For example, if a participant missed the whole snacks section, were they excluded or were all responses were imputed as zero?
Reading on to results I can see this is addressed, but suggest that the unsatisfactory response explanation be moved to methods.
Results
Table 1: SD, gr and ml also require footnote/abbreviations explanation
Tables 2 and 3: Consistent terminology (Ref versus reference)
Figure 1: What is the scale on the x axis, and why is higher reward on left and lower on right? This could be confusing for the reader, but may have an explanation? It is evident from the methods that this is a continuous variable, but from the figure only, could look like categorical and therefore be confusing given the type of graph.
Discussion
Are large proportion of the first section (two paragraphs) of the discussion restates the results without contextualizing the findings within the existing literature. This section needs to be updated so that each statement or finding is referenced against literature or towards future work or has some explanatory information for context around the finding. It might be that some of the limitations stated can be used for this, rather than being stated all together?
The conclusion that has been added also restates the results, and does not add any further direction, except in final sentences. Suggest more sentences such as the following for situating the findings and providing clarity around the direction and justification for future research.
The finding that…….indicates that…….. is needed to further investigate……..
Author Response
see attach
